# CRISPR/Cas9-Mediated Knockout of the White Gene in *Agasicles hygrophila*

**DOI:** 10.3390/ijms26104586

**Published:** 2025-05-10

**Authors:** Li Fu, Penghui Li, Zhiyi Rui, Jiang Sun, Jun Yang, Yuanxin Wang, Dong Jia, Jun Hu, Xianchun Li, Ruiyan Ma

**Affiliations:** 1College of Plant Protection, Shanxi Agricultural University, Jinzhong 030801, China; fuli200940810105@163.com (L.F.); lphkeyan@163.com (P.L.); ruizyama@126.com (Z.R.); sunjiang198@163.com (J.S.); yangjuncau@163.com (J.Y.); wangyuanx1992@163.com (Y.W.); biodong@foxmail.com (D.J.); hujun@sxau.edu.cn (J.H.); 2Department of Entomology and BIO5 Institute, University of Arizona, Tucson, AZ 85721, USA

**Keywords:** CRISPR/Cas9, gene editing, *Agasicles hygrophila*, white gene

## Abstract

*Agasicles hygrophila* is the most effective natural enemy for the control of the invasive weed *Alternanthera philoxeroides* (Mart.) Griseb. However, research on the gene function and potential genetic improvement of *A. hygrophila* is limited due to a lack of effective genetic tools. In this study, we employed the *A. hygrophila white* (*AhW*) gene as a target gene to develop a CRISPR/Cas9-based gene editing method applicable to *A. hygrophila*. We showed that injection of Cas9/sgRNA ribonucleoprotein complexes (RNPs) of the *AhW* gene into pre-blastoderm eggs induced genetic insertion and deletion mutations, leading to white eyes. Our results demonstrate that CRISPR/Cas9-mediated gene editing is possible in *A. hygrophila*, offering a valuable tool for studies of functional genomics and genetic improvement of *A. hygrophila*, which could potentially lead to more effective control of invasive weeds through the development of improved strains of this biocontrol agent. In addition, the white-eyed mutant strain we developed could potentially be useful for other transgenic research studies on this species.

## 1. Introduction

CRISPR/Cas9 (Clustered Regularly Interspaced Palindromic Repeats, CRISPR) is a newly discovered, powerful genome editing tool that is active in all organisms [1,2]. In this editing system, the *Streptococcus pyogenes* Cas9 protein, guided by an RNA endonuclease, precisely cuts a double-stranded DNA based on base paring between CRISPR RNA (crRNA) and the target sequence (protospacer). The double-stranded DNA breaks (DSBs) are then repaired either via non-homologous end joining (NHEJ) or homology-directed recombination (HDR) in the presence of a donor DNA template sequence. NHEJ repairs are of low fidelity and can either add or subtract bases to produce protein mutations or frameshift mutations [3]. CRISPR-based genome engineering has been implemented in many insect orders, but in the order Coleoptera, it has been carried out in only a few species (i.e., *Tribolium castaneum* [4], *Leptinotarsa decemlineata* [5], *Diabrotica virgifera virgifera* [6], and *Harmonia axyridis* [7]).

As an easily visible marker, eye color has frequently been manipulated genetically to induce genetic transformations in various types of insects. The eye color of *Drosophila melanogaster* is determined by ommochrome and pteridine pigments. The *white* (*Dm-w*) gene encodes an ABC transporter, which is essential for transporting ommochrome and pteridine precursors across membranes [8,9]. The *Dm-w* mutation results in white compound eyes [10]. White forms heterodimers with brown (bw) and scarlet (st), which are responsible for the transport of pteridine precursors and ommochromes, respectively. The *bw* and *st* mutants have brown [11] and scarlet eyes [12]. As it is visible and does not affect normal growth, the artificially optimized form of *Dm-w*, called mini-white [13], is the most commonly used marker for site-specific recombinases that mediate genome insertion, such as the FLP/FRT system [14], the ΦC31-mediated recombinase cassette exchange (RMCE) system [15], and the Cre-mediated recombination system [16]. In these systems, the *Dm-w* mutant strain is used as the starting strain and the donor plasmid contains *mini-W*. This transformation changes the eye color from white back to red, which is the normal eye color in *Drosophila* species. The greatest advantage of using this system is that eye color functions as a visible screening marker, thus avoiding the need to use fluorescent proteins. The *white* gene is conserved in all insects, and knockdown expression or knockout results in white eyes [17,18,19]. Therefore, this system can be used for non-model insect species.

The flea beetle *Agasicles hygrophila* (Selman and Vogt), which originated from Argentina, has been introduced into various countries, including China, to successfully control the invasive aquatic weed *Alternanthera philoxeroides* (Mart.) Griseb [20]. RNA interference (RNAi) is the primary tool used to study gene function in various organisms, including *A. hygrophila*. However, although RNAi can be efficiently applied in *A. hygrophila* [21], this approach is limited to knockdown expression for partial reduction at the transcriptional level, and its use results in only a limited time period of knockdown. More efficient gene knockout methods are needed for use in *A. hygrophila*. Using the CRISPR/Cas9-mediated germline transformation technique, we employed the *white* gene as a visible marker to produce mutant forms of *A. hygrophila.* In this article, we describe a simple and efficient approach for genome editing in *A. hygrophila* using the CRISPR/Cas9 system.

## 2. Results

### 2.1. Phylogenetic Identification of the A. hygrophila White (AhW) Gene

A reciprocal TBLASTN search and a phylogenetic analysis were combined to identify the *A. hygrophila white* gene. The TBLASTN search of the *A. hygrophila* transcriptome dataset using the *D. melanogaster* white protein (Dm-w, GenBank accession # CAA26716.2) as the query found that the best hit was *A. hygrophila* unigene # c3849/f1p0/2312 (e value = 0, total score = 627, and query coverage = 88%), which shared an amino acid sequence identity of 53% with the *D. melanogaster* white protein gene. The amino acid similarity of the *D. melanogaster* white protein gene to other *A. hygrophila* unigenes was less than 30%. A Blastp search of the *D. melanogaster* gene dataset, with the translated *A*. *hygrophila* unigene # c3849/f1p0/2312 as the query, revealed that the best hit for *A. hygrophila* unigene # c3849/f1p0/2312 was the *D. melanogaster* white protein (Table 1; e value = 0, total score = 602, query coverage = 100%, and identity = 49.35%). Such a one-to-one orthologous relationship between the *D. melanogaster* white gene and *A. hygrophila* unigene # c3849/f1p0/2312 was consistent with our phylogenic analysis, which clearly showed that *A*. *hygrophila* unigene # c3849/f1p0/2312 clustered with other previously identified insect white protein genes from the orders Coleoptera, Diptera, and Lepidoptera, and that it was more distantly related to the scarlet and brown protein genes (Figure 1). These data confirm that *A. hygrophila* unigene # c3849/f1p0/2312 represents the transcript of the *A. hygrophila* white gene (*AhW*).

The full-length cDNA sequence of *AhW* was amplified with the primers AhW-F (located upstream of the putative start codon initiator) and AhW-R (located downstream of the putative stop codon) (Appendix A) and then submitted to GenBank (accession number OR123872). Alignment of the obtained 2309 bp *AhW* cDNA with the unpublished *A. hygrophila* genome (personal communication with Dr. FangHao Wan) showed that the genomic DNA sequence of *AhW* is over 4917 bp long and comprises 12 exons and 11 introns (Figure 2A). The translated protein sequence of the *AhW* cDNA is 669 amino acids in length and exhibits the typical characteristics of the ABC transporter G subfamily, which include a C-terminal transmembrane domain (TMD) and a nucleotide-binding domain (NBD) composed of a Walker A motif, an ABC signature motif, a Walker B motif, a Q loop, a D loop, and an H loop (Figure 2B).

### 2.2. Creation of an AhW Knockout White-Eyed Strain

sgRNA1 and sgRNA2 are respectively located in exons 3 and 5 of the *AhW*. A total of 300 eggs (G0) were injected with RNP, consisting of Cas9 protein, sgRNA1, and sgRNA2. In the end, we obtained eight adults (three females and five males), with only one mutant female, which had two white and black chimera eyes (Figure 3A). The right eye of this G0 mutant female was mostly white, with a small number of black facets at the bottom edge, while its left eye was composed of three black faceted stripes and two white faceted stripes (Figure 3A).

To confirm that the *AhW* gene sequence was indeed successfully edited, we extracted the genomic DNA of the wild-type strain (AhW strain) beetle and the G0 white-eyed mutant and then amplified the sequences of the *AhW* editing sites. Electrophoretic results showed that the wild-type beetle’s *AhW* had a main band of 1818 bp. However, besides this main band, there were several shallow bands smaller than 1818 bp in the G0 white-eyed beetle (Figure 4).

To establish a homozygous white-eyed mutant strain, the G0 mutant female was allowed to mate with two wild-type black-eyed males. In the resultant G1 progeny, all 11 females had black eyes, whereas all males (8 in total) had white eyes. The eight white-eyed G1 males were allowed to mate with eight black-eyed G1 females in eight separate mating cages; all eight cages produced G2 progeny in which both females and males were ~50% white-eyed and ~50% black-eyed (a G2 mutant male and a female are shown in Figure 3B). This suggests that G0 *AhW* mutations responsible for white eyes in the *AhW* knockout strain (AhW-KO) are likely homozygous and fixed.

### 2.3. Mutations in the Two sgRNA Target Sites in AhW gDNA from the AhW-KO Strain

To identify the mutations corresponding to the two sgRNA target sites located in exons 3 and 5 of *Ahw* (Figure 2A), we used the genomic DNA samples individually extracted from one hind leg of each of the eight G_1_ white-eyed males as the templates to PCR-amplify the regions flanking the sgRNA target sites (Appendix A) using the gene-specific primers AhW-F1 and AhW-R1 (Appendix A). Direct sequencing of the PCR products from each of the eight G_1_ white-eyed males showed that all eight PCR products had one unique disruptive mutation (Figure 5A). These included one 800 bp deletion spanning from the sgRNA1 target site to the sgRNA2 target site, one −1 bp/+2 bp indel and three deletions (−7, −1, and −2 bp) corresponding to the sgRNA1 target site, and two −1 and −4 bp deletions and one +4 bp insertion corresponding to the sgRNA2 target site (Figure 5A). The −1 bp/+2 bp indel in AhW-KO5 directly truncated the AhW protein with a premature stop codon, whereas all the other seven CRISPR/Cas9-induced mutations disrupted the *AhW* coding sequence by first shifting the reading frame, followed by the introduction of premature stop codons at various positions (Figure 5B).

### 2.4. Inheritance of Cas9-Introduced White Eye Mutations

Since all the males of the G1 generation obtained by mating with the wild type exhibited white eyes, while all the females exhibited black eyes, we speculate that *AhW* recessive inheritance of X chromosome. To test this speculation, two reciprocal single-pair crosses between the wild-type AhW strain and the AhW-KO strain (1 ♀AhW × 1 ♂AhW-KO; 1 ♂AhW × 1 ♀AhW-KO) were made to yield G1 and G1′ progeny, respectively, and all males and females of both G1 and G1′ hybrids were allowed to self-cross to produce G2 and G2′ generations. All the G1 hybrids and G2 progeny were individually examined for their sex and eye color phenotype. As shown in Table 2, all G1 hybrids, regardless of sex, had black eyes, while 50% of the G1′ hybrids had white eyes, and we observed only white eyes for males and black eyes for females. In addition, the black/white eye ratio of the self-inbred G2 offspring was 2.92:1.0 (no difference from 3:1), with all females and about half of the males having black eyes and the other half of the males having white eyes. For comparison, the black/white eye ratio was 1:1 for the self-inbred G2′ generation, regardless of sex (Table 2). Such an inheritance pattern demonstrates that the Cas9-introduced white eye mutation is an X-linked recessive trait that affects males more than females.

### 2.5. Time Course of AhW Expression and Blackening of Eyes

The compound eyes of the AhW strain were colorless and transparent on the day of pupation. They turned light red on the second day, gradually turned black on the third and fourth days, and remained black during the rest of the pupal stage (Figure 6) and throughout adult life (Figure 3B). In contrast, the eyes of the AhW-KO mutant strain were colorless throughout the pupal and adult stages (Figure 6). Similar to the time course for eye blackening, in the AhW strain, the *AhW* expression level was not different in the first three days of the pupal stage but rapidly reached a peak on the fourth day. It then dropped to its lowest level on the fifth day (Figure 7).

## 3. Discussion

Because eye color traits are easy to observe, do not require any professional equipment, and are free of fluorescent proteins, they are excellent markers for identifying germline transformation and gene editing events in insects. We identified the *A. hygrophila* homolog of the *white* gene, which has been previously reported as important in ommochrome synthesis for some fly species and the red flour beetle (*T. castaneum*). We characterized the mutant phenotype of this gene in *A. hygrophila* using the CRISPR/Cas9 system. By injecting the cas9 protein/gRNA complex into early *A. hygrophila* embryos, we achieved high rates of germline transformation. Our work validates that the CRISPR/Cas9 system is highly active in *A. hygrophila* for generating site-specific mutations. The ommochrome synthesis pathway of the model species *T. castaneum*, whose eye pigmentation is solely determined by ommochromes, has been well studied. There are three key genes involved in the ommochrome synthesis pathway: (1) the tryptophan oxidase gene *vermilion* (*ver*), which oxidizes the upstream component tryptophan to formylkynurenine; (2) ABC transporters such as *White*, which transfer 3-hydroxykynurenine into pigment granules; and (3) *Cardinal*, which auto-oxidizes 3-hydroxykynurenine to xanthommatin. Any functional abnormality in these genes leads to white eye phenotypes [17,22,23].

In addition to its involvement in ommochrome synthesis, *White* may have other functions. Knockout of *White* was lethal in *Helicoverpa armigera* and *Oncopeltus fasciatus* [18,24], and led to complete failure of copulation in *Drosophila suzukii* [25]. These phenotypes may arise from a loss of function, but it cannot be ruled out that they may be caused by cas9 off-targeting.

We injected Cas9 protein/gRNA into *A. hygrophila* embryos to achieve gene editing. This method is currently used for gene editing in most insects, which requires eggs to be injected individually. Due to the physical damage caused by injection, many eggs die during embryogenesis, and a large number of eggs must be injected to obtain a sufficient number of viable transformed adults. This greatly increases the work required. Recently, it has been reported that injection of the Cas9 protein/gRNA into the female hemocoel causes G1 eggs to undergo gene editing, and this strategy has also been shown to be effective in other beetles [22,26,27]. This approach has several advantages. First, injecting adults is significantly easier than injecting eggs. Second, adult injection is less likely to cause mechanical damage resulting in mortality. Third, since each *A. hygrophila* female can lay 20–30 eggs at a time, injecting a small number of females could produce a large number of transformed offspring, thereby greatly reducing the workload. The feasibility and effectiveness of this alternative injection process in *A. hygrophila* should be confirmed in future research.

The *white* gene and *white* mutants identified in this study are ideal materials for gene function research. As transgenic materials obtained from studies of *Drosophila* [14,15,16], the *white* gene and *white* mutants can be used as selectable markers for RMCE systems and provide useful genetic materials for the study of *A. hygrophila*. In addition, the 3xP3-EGFP is another transformation marker used in a variety of insects, including those in the order Coleoptera. However, the fluorescence of the eye is very weak in the wild type and is not easy to observe, but it is quite bright and easy to screen for in light-colored mutants, such as those with white or red eyes [28,29]. Therefore, the AhW-KO white eye line obtained in this study can be used for genetic engineering, and 3xP3-EGFP can be used as a screening marker. In fact, the *white* mutant (*w*^1118^) is the material most used to generate P-element insertion lines, including the widely-used GAL4/UAS system in *Drosophila* studies [30].

Because they do not contain foreign genes, gene editing products are subject to limited regulatory supervision. We have established a highly efficient gene knockout system in *A. hygrophila*, and this system and *AhW* mutants should be beneficial for the study of molecular interactions between weeds and their natural enemies as biological control agents, as well as for developing new strains with better ecological adaptability and control effects.

## 4. Materials and Methods

### 4.1. Insects

The *A. hygrophila* individuals used in this work were originally collected near the South China Agricultural University (Guangzhou, China), and a colony was reared in a growth chamber at 25–28 °C, with 85 ± 5% rh and a 14:10 L:D photoperiod. The larvae and adults were fed with fresh *A. philoxeroides* foliage, which was initially collected from Yuhuan County, Zhejiang Province, and thereafter grown for this use in a greenhouse at Shanxi Agricultural University.

### 4.2. Sequence Amplification and Analysis

Total RNA was isolated from *A. hygrophila* adults using RNAiso Plus (Takara, Kusatsu, Japan) according to the manufacturer’s protocol. cDNA was synthesized from 1 µg of total RNA with random primers using M-MLV Reverse Transcriptase (Promega, Madison, WI, USA). The open reading frames (ORFs) of *AhW* were amplified using Phanta Super-Fidelity DNA Polymerase (Vazyme, Beijing, China) with the primer of AhW-F/R (Appendix A). The PCR products were subcloned using an M5 HiPer pTOPO-Blunt Cloning Kit (Mei5bio, Beijing, China) and sequenced by Sangon (Shanghai, China). To analyze the evolutionary relationship between AhW and eye pigment synthesis genes, the protein sequences of the *white*, *scarlet*, and *brown* were used to construct an evolutionary tree. The phylogenetic tree was constructed using the Neighbor-Joining method with MEGA (ver. 11.0.13) [31]. Numbers in the tree indicate bootstrap values (1000 replicates). DeepTMHMM (Transmembrane Helices Hidden Markov Models) was used for transmembrane prediction (https://dtu.biolib.com/DeepTMHMM) (16 March 2020).

### 4.3. sgRNA Design and Synthesis

The sgRNAs (single-guide RNAs) were designed using CRISPOR (http://crispor.tefor.net/crispor.py) (8 April 2020) [32]. To avoid the possibility of off-targeting, we performed a local blast (BlastN) comparison of the designed gRNAs with the *A. hygrophila* transcriptome [33] and the non-published genome to ensure that there were no consecutive 8-base matches. Mismatches 8 to 14 bp upstream of the PAM cause off-targeting [34].

sgRNAs with high scores and specificity were selected. The sgRNAs were prepared according to Kistler et al. [35]. The sgRNA templates were synthesized using template-free PCR with two oligos. The forward primer (Appendix A) comprised a T7 promoter, an sgRNA sequence, and part of the sgRNA scaffold. The reverse primer contained a partial sgRNA scaffold. The sgRNA templates were amplified with Phanta Super-Fidelity DNA Polymerase (Vazyme), and the PCR products were purified using the E.Z.N.A Gel Extraction Kit (Omega, Biel/Bienne, Switzerland). Then, we used the purified products for in vitro transcription and purification using the T7 Ribomax Express RNAi System (Promega), following the manufacturer’s instructions. The sgRNAs were determined using a NanoDrop 2000 spectrophotometer (Thermo Fisher, Waltham, MA, USA), diluted to 3 µg/µL, and stored at −80 °C until use.

### 4.4. Embryonic Microinjection

The Cas9/sgRNA complex was produced by mixing 300 ng/μL sgRNA, 300 ng/μL Cas9 protein (NEB), 0.1% Phenol Red, and 1× NEBuffer r3.1; the mixture was incubated for 10 min at room temperature to form RNPs, and then the tubes containing the mixture were kept on ice. The eggs of *A. philoxeroides*, which were laid within the previous 30 min on fresh leaf discs, were used for microinjection using the FemtoJet 4× microinjection system (Eppendorf, Pune, Maharashtra). Since the eggs of the *A. hygrophila* were laid in two neat rows on the leaves of *A. philoxeroides* (Appendix A), there was no need to align the eggs neatly to facilitate microinjection. The needles used in this procedure were prepared from glass capillaries (1.0 mm on the outside and 0.5 mm on the inside; VitalSense Scientific Instruments, Wuhan, China) using a P-97 micropipette puller (Sutter Instrument, Novato, CA, USA; P: 500, heat: 500, pull: 60, vel: 75, time: 90). The needles were inserted from the posterior of the eggs to minimize mechanical damage, and the injection was completed within 30 min. The injected eggs were then placed in glass Petri dishes covered with moist filter papers and kept in a growth chamber.

### 4.5. Mutagenesis Analysis

To select germline-transmitted mutations, emerged G0 adults showing white eyes were screened. Then, G0 progeny were mated with the wild type individually to acquire G1 progeny. Genomic DNA was extracted from single beetles by homogenizing the beetles in 50 µL of the squishing buffer (25 mM NaCl; 1 mM EDTA; 10 mM Tris-HCl, pH 8.2; and 200 µg/mL of proteinase K). After heating to 37 °C for 30 min, followed by inactivation at 95 °C for 2 min, the PCR products spanning all sgRNA target sites were amplified using specific primers (Appendix A). PCR was performed using Phanta Super-Fidelity DNA Polymerase (Vazyme). The PCR conditions consisted of 95 °C for 3 min, followed by 35 cycles of 95 °C for 15 s, 55 °C for 15 s, and 72 °C for 2 min 20 s, a final extension step at 72 °C for 5 min, and holding at 4 °C. The products were gel-purified for direct sequencing at Sangon Biotech (Shanghai, China) to determine the exact indel type.

### 4.6. Reverse Transcription Quantitative PCR (RT-qPCR)

To investigate the spatiotemporal distribution of the white gene, total RNA was isolated from pupae (1–5 days old) using RNAiso Plus (Takara). The RNA samples (1 µg each) were used to synthesize the first-strand cDNA using the HiScript III 1st Strand cDNA Synthesis Kit (+gDNA wiper) (Vazyme, Nanjing, China). Reverse transcription quantitative PCR (RT-qPCR) was carried out using ABI 7500 (Applied Biosystems, Foster City, CA, USA) with the ChamQ Universal SYBR qPCR Master Mix (Vazyme). Amplifications were performed in 20 µL reaction volumes consisting of 10 µL of the mix, 0.8 µL of each primer (10 µM), 7.4 µL of nuclease-free water, and 1 µL of cDNA (0.05 µg/µL). The PCR conditions consisted of 95 °C for 3 min, followed by 42 cycles of 95 °C for 10 s, 55 °C for 15 s, and 72 °C for 30 s. The melt curves were used to assess the specificity of amplification. The ribosomal protein gene S18 (*Rps18*) was used as an internal reference.

### 4.7. Eye Pigment Observation

Eye pigment observation of the adults and pupae from the first to the fourth day was performed under a Leica M205C stereomicroscope (Leica Microsystems, Wetzlar, Germany) with bright-field filters. To prevent them from moving under the microscope, the adults were frozen with liquid nitrogen, but the pupae were not treated.

### 4.8. Statistical Analysis

One-way ANOVA, followed by Tukey’s honest significant difference (HSD) test at *p* < 0.05, was used to detect significant differences among the means of pupal durations across the treatments using SPSS 22.0 software (IBM, Armonk, NY, USA).

## Figures and Tables

**Figure 1 ijms-26-04586-f001:**
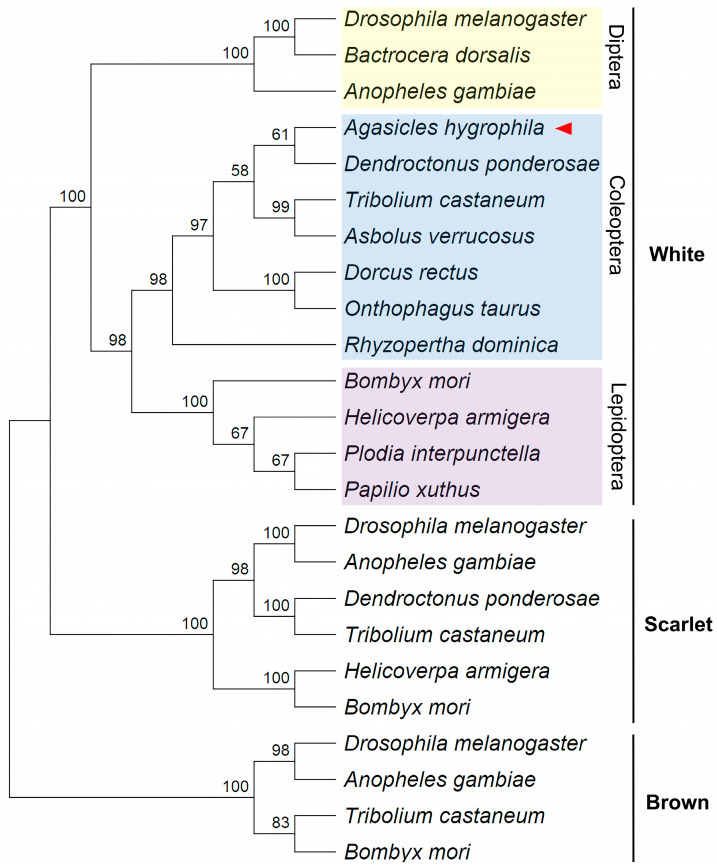
Phylogenetic analysis of the white, scarlet, and brown genes among insects. The tree was constructed based on the deduced amino acid sequences using the Neighbor-Joining method. The numbers in the tree indicate the bootstrap values (1000 replicates). The sequences were retrieved from the GenBank database, and their accession numbers are shown in Appendix A.

**Figure 2 ijms-26-04586-f002:**
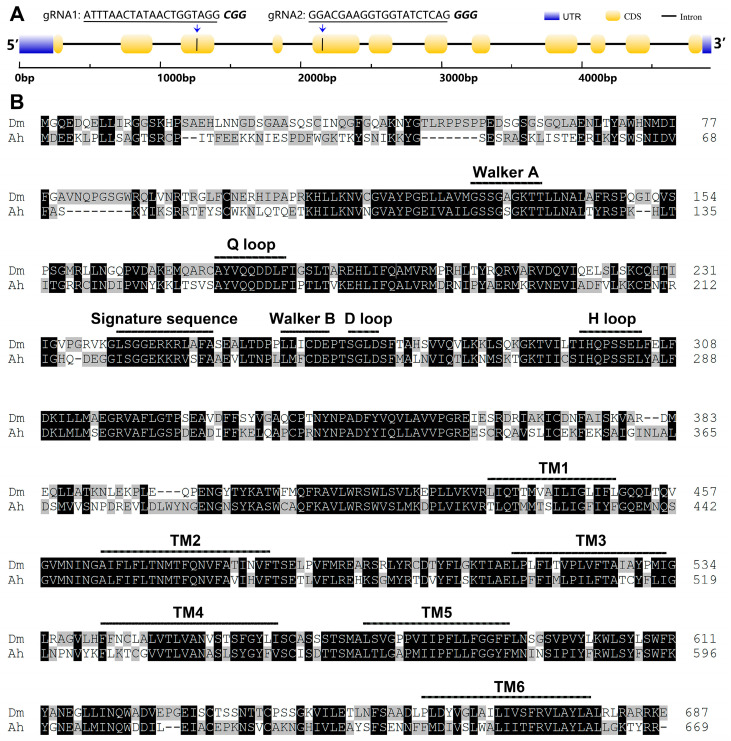
Schematic diagram of the gene structure, protein transmembrane region, and conserved domain of *AhW*. (**A**) The genome of *AhW* consists of 12 exons (yellow boxes) spanning 4917 bp. The crRNA targets, gRNA1 and gRNA2, are located in exons 3 and 5, respectively, and are shown with underlining and protospacer adjacent motifs (PAMs) (in bold italics). (**B**) Comparative alignments of the white protein sequences of *D. melanogaster* and *A. hygrophila*: (1) TM1-TM6 are 6 transmembrane helices; (2) the Walker B motif (hhhhDE): IMFCDE for beetle and LLICDE for fly; (3) the D loop (SALD): SGLD for both insects; (4) the Q loop: AYVQQDDLFi for both insects; and (5) the H loop: IHQPSSEL for both insects. Dm for *D. melanogaster*, Ah for *A. hygrophila*.

**Figure 3 ijms-26-04586-f003:**
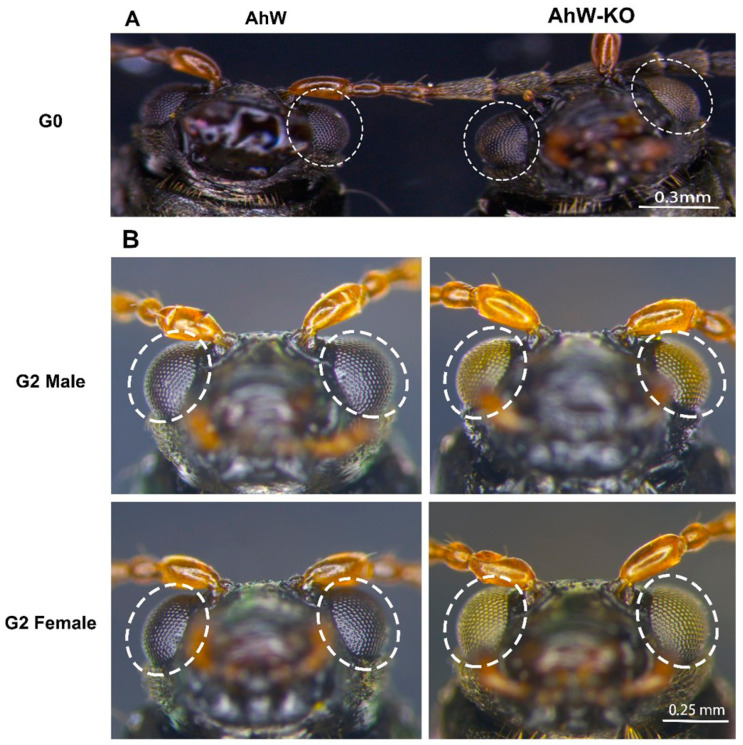
Photographs of wild-type AhW and AhW-KO mutant eyes. Somatic CRISPR edits of G0 adults (**A**) and germline-based G2 progeny (**B**).

**Figure 4 ijms-26-04586-f004:**
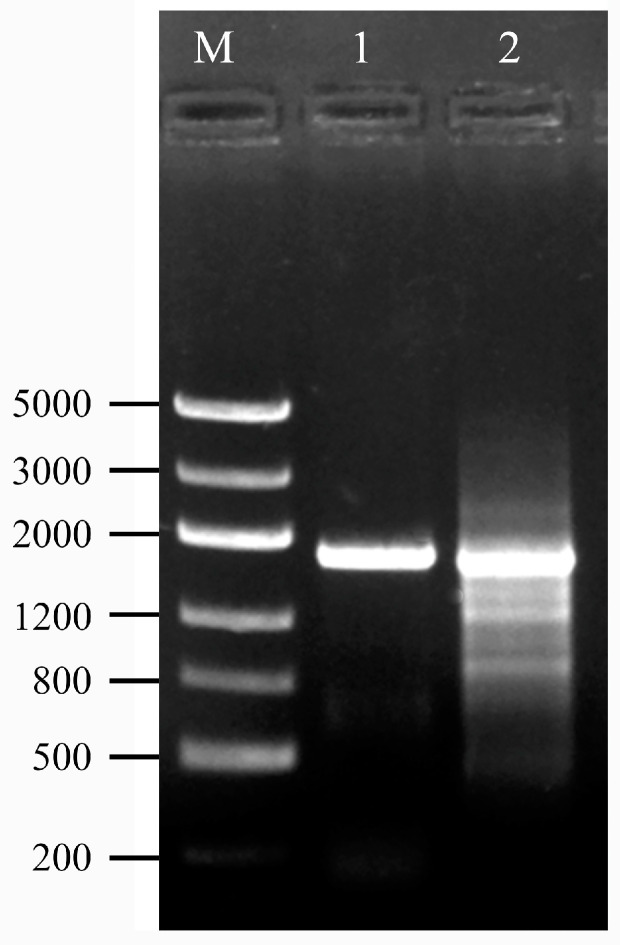
Electrophoresis of *AhW* editing site sequences. The genomic DNA of wild-type and G0 white-eye beetles was extracted, and then the sequences of the *AhW* editing sites were amplified and electrophoresized. M, DNA marker; lane 1, *AhW* of wild type; lane 2, AhW-KO of G0 white-eyed mutant.

**Figure 5 ijms-26-04586-f005:**
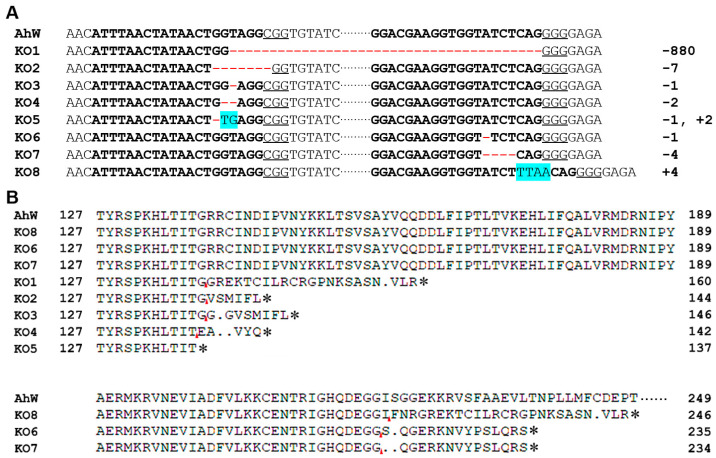
CRISPR-edited AhW-KO sequences of mutants. (**A**) Multiple types of deletions and/or additions cause frameshifts. The sgRNA target sequence is labeled in bold in the WT reference sequence, with PAMs underlined. Blue backgrounds indicate insertions, and red short horizontal lines indicate deletions; the long solid red line indicates a large missing fragment. (**B**) Protein sequences of the AhW-KO mutants: KO1-KO8 are AhW knockout mutants 1–8, where the red arrows indicate a frameshift mutation site. Asterisks (*) mark the stop codons, and ellipses stand for subsequent amino acid sequences that are not listed.

**Figure 6 ijms-26-04586-f006:**
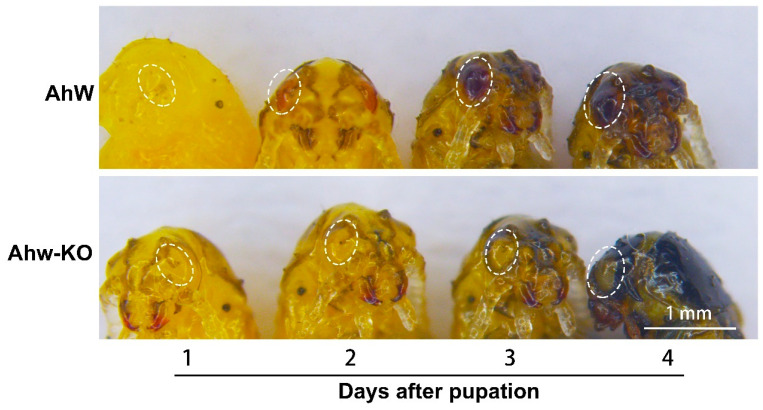
Photographs of pupal eyes of AhW strain beetles and AhW-KO mutants. The ellipses mark the eye regions.

**Figure 7 ijms-26-04586-f007:**
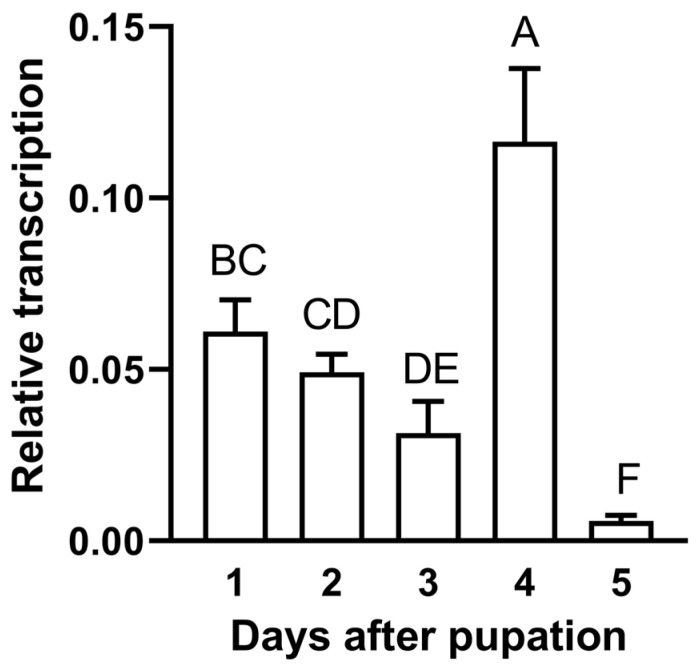
Expression profiles of *AhW* gene in pupae aged 1–5 days. Data analysis was based on one-way ANOVA and Tukey’s honest significant difference (HSD) tests, Values in parentheses indicate 95% confidence intervals. A–F: significant difference, *p* < 0.05. Each developmental stage was examined with three biological replicates.

**Table 1 ijms-26-04586-t001:** Protein sequence search results for AhW with *D. melanogaster*.

Gene	Total Score	Query Covery	E Value	Identity
Dm-white	694	100%	0	49.35%
Dm-scarlet	399	85%	5 × 10^−130^	36.12%
Dm-brown	250	86%	2 × 10^−73^	27.26%

**Table 2 ijms-26-04586-t002:** Two single-pair reciprocal crosses and the black/white eye ratios of the male and female progeny in the G1 and G2 generations.

Generations	Parents	Total Adult Progeny	Female Progeny	Male Progeny	Females with Black Eyes	Females with White Eyes	Males with Black Eyes	Males with White Eyes	Black/White Ratio for All Progeny	Black/White Ratio for Females	Black/White Ratio for Males
G1 hybrid	1 ♀AhW × 1 ♂AhW-KO	83	44	39	44	0	39	0	1.0:0	1.0:0	1.0:0
G1′ hybrid	1 ♀AhW-KO × 1 ♂AhW	72	38	34	38	0	0	**34**	**1.1:1**	**1:0**	**0:1.0**
Self-inbred G2	44 ♀G1 × 39 ♂ G1	486	232	254	232	0	130	**124**	**2.92:1**	**1:0**	**1.05:1.0**
Self-inbred G2′	38 ♀G1′ × 34 ♂ G1′	433	226	207	115	**111**	108	**99**	**1.06:1.0**	**1.04:1.0**	**1.09:1.0**

## Data Availability

The original contributions presented in this study are included in the article/Appendix A. Further inquiries can be directed to the corresponding author.

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
