# Peer review of "CRISPR/Cas9-Mediated Knockout of the White Gene in Agasicles hygrophila"

_ijms, 2025, doi:10.3390/ijms26104586_

Round 1
Reviewer 1 Report
Comments and Suggestions for Authors
The manuscript "CRISPR/Cas9-mediated knockout of the white Gene in Agasicles hygrophila" is a study focused on developing a genetic editing by using the CRISPR/Cas9 system. The current study uses the white gene as a marker of success in genetic manipulations, as is widely used in Drosophila.
Although all results are clear, and gene editing was successful, I have an observation that needs to be clarified before this work is published.
In the Figure 3A, AhW-KO, its observed a mosaic phenotype, which is an indication of somatics edtion instead of a germinal edition. This is in agreement with the Figure 4. The author should analyze the genomic DNA PCR of all mutants throughout the generation to ensure the mosaic phenotype disappears during the crosses.
Later in Figure 5, it is not clear if all the mutations were intentional or unexpected. Author have to clarify this. This results are also in agreement with mosaic phenotype.
-Line 193 says Figure 1A, Do the authors wanted to say Figure 2A?
Author Response
Comments and Suggestions for Authors
The manuscript "CRISPR/Cas9-mediated knockout of the white Gene in Agasicles hygrophila" is a study focused on developing a genetic editing by using the CRISPR/Cas9 system. The current study uses the white gene as a marker of success in genetic manipulations, as is widely used in Drosophila.
Although all results are clear, and gene editing was successful, I have an observation that needs to be clarified before this work is published.
In the Figure 3A, AhW-KO, its observed a mosaic phenotype, which is an indication of somatics edtion instead of a germinal edition. This is in agreement with the Figure 4. The author should analyze the genomic DNA PCR of all mutants throughout the generation to ensure the mosaic phenotype disappears during the crosses.
Response: According to the suggestion, we have added the agarose gel electrophoresis images of the genomic DNA PCR products of all mutants to the supplementary materials (Fig. S2).
Figure S2. Electrophoresis of AhW editing site sequences of G1. The genomic DNA of wild-type and G1 white-eye male beetle was extracted, and then the sequences of the AhW editing sites were amplified and electrophoresized. M, DNA marker; lane W wild-type, lane 1-8, AhW KO1-KO8 of G1 white-eyed mutant.
Later in Figure 5, it is not clear if all the mutations were intentional or unexpected. Author have to clarify this. This results are also in agreement with mosaic phenotype.
Response: All the mutations were expected as all the mutations occurred near PAM, and all the mutants sequenced were white-eyed.
-Line 193 says Figure 1A, Do the authors wanted to say Figure 2A?
Response: Thanks for pointing out the error, the mistakes have been corrected. Furthermore, we also checked all the other legends to ensure consistency with the content in the manuscript.
Reviewer 2 Report
Comments and Suggestions for Authors
The authors provide a report on the first-time use of CRISPR in the coleopteran insect species, Agasicles hygrophila. In this study, the white gene was knocked out and the genetics of the knockout were characterized. Minor revisions are required before this manuscript would be suitable for publication. Comments are provided for each section.
Introduction.
Line 42-43. “The artificially optimized form of Dm-w called mini-white”
Since this is an artificial construct, a citation is needed here for the creation or first use of this artificially optimized form.
Materials and Methods.
Line 79. “phylogenetic tree were constructed”
A phylogenetic tree consisting of what? Please, here, indicate the scope of the analysis. Some relevant information is found in the legend of Figure 1 and in Table S2, but more details about the approach and what was analyzed needs to be mentioned here.
Line 92-93. “The forward primer (Table S1) comprised a T7 promoter… The reverse primer contained a partial sgRNA scaffold…”
In Table S1, for the sgRNA template primers it would be helpful to the reader to demarcate which parts of the sequence represent the T7 promoter, the sgRNA sequence and the sgRNA scaffold respectively. This could be done using bold font and/or different colored fonts, with a footnote to clarify the different features.
Line 102. “Cas9”
Is this referring to Cas9 protein or Cas9 mRNA? Please clarify here.
Line 106-107. “no further tagging was necessary.”
What is meant by tagging? This is not clear.
Line 120-121. “the PCR products spanning all sgRNA target sites were amplified using specific primers”
Information about PCR thermocycling conditions is needed here.
Results.
Line 192-193. “A total of 300 eggs (G0) were injected with CRISPR-Cas9 ribonucleoprotein mixtures containing sgRNA1 and sgRNA2.”
For sake of clarity, please indicate if sgRNA1 and sgRNA2 were injected together in one mixture with Cas9, or if these were injected in separate RNP mixtures.
Line 200. “genomes”
Replace with genomic DNA
Line 231. “AhW-KO_m 1”
Please verify and correct this as appropriate. In Figure 5, the only -1/+2 indel is seen for KO-5, not KO-1, which shows the 800bp deletion.
Line 239-240. “the wild-type AhW strain and the AhW-KO strain”
This is confusing, to refer to the wild type strain as AhW and the mutant strain as AhW-KO. Please consider to call the wild-type strain something more distinct.
Line 255-256. “and remained black during the rest of the pupal stage and throughout adult life (Figure 6)”
Does Figure 6 show the adult stage? The figure legend for Figure 6 says only pupal eyes, in that case, it’s not clear how anything can be mentioned about the adult stage here.
Discussion.
Line 283-284. “homologues of the white genes”
It is unclear what is being referred to here in the plural. Isn’t the target gene in this study only one homologue of one white gene?
Line 285-286.
“We characterized the mutant phenotype of these genes”
Again, here, isn’t it only one gene that you characterized the mutant phenotype for?
Line 288. “high rates of germline transformation”
From lines 192-194, it was indicated that 300 embryos were injected, from this, 8 adults were obtained, and only one showed evidence of the mutant phenotype. Considering these results, it is not clear what is meant by high rates of germline transformation.
Line 317. “As transgenic materials obtained from studies of Drosophila”
The construction of this sentence is confusing. What is meant by “as transgenic materials obtained from studies of Drosophila”? What transgenic materials have been obtained for this manuscript?
Line 329. “We have established a highly efficient gene knockout system”
Referring back to my previous comment about the high rates of germline transmission, I think it is premature to say that it is highly efficient in this species based on the data provided here. It would be more accurate to simply say that you confirmed the efficacy of CRISPR-CAS9 in this species, A. hygrophila. This is a very nice result, but it should be overstated at this point without the corresponding data.
Figures and Tables.
Figure 1. Since the scarlet and brown genes are included here in this figure, there should be some mention of these genes in the introduction, most appropriately where the white gene is being described.
Figure 3. For Figure 3B, it is not clear what we are looking at. Does the AhW and AhW-KO headers for 3A also apply to 3B? Furthermore, in the figure legend, on line 214, it mentioned “and germline-based G1 progeny (B). However, in the figure for the 3B panels, it says G2 male and G2 female, not G1 insects.
Comments on the Quality of English LanguageOverall, the manuscript is very readable, though several minor errors in the language are found throughout the manuscript.
Author Response
The authors provide a report on the first-time use of CRISPR in the coleopteran insect species, Agasicles hygrophila. In this study, the white gene was knocked out and the genetics of the knockout were characterized. Minor revisions are required before this manuscript would be suitable for publication. Comments are provided for each section.
Introduction.
Line 42-43. “The artificially optimized form of Dm-w called mini-white”
Since this is an artificial construct, a citation is needed here for the creation or first use of this artificially optimized form.
Response: Reference 11 (Sun Y.H., et al. White as a reporter gene to detect transcriptional silencers specifying position-specific gene expression during Drosophila melanogaster eye development. Genetics 1995, 141, 1075-1086.) have been added in line 44.
Materials and Methods.
Line 79. “phylogenetic tree were constructed”
A phylogenetic tree consisting of what? Please, here, indicate the scope of the analysis. Some relevant information is found in the legend of Figure 1 and in Table S2, but more details about the approach and what was analyzed needs to be mentioned here.
Response: The following content has been added in line 80: To analyze the evolutionary relationship between AhW and eye pigment synthesis gene in other insects, the protein sequences of white, scarlet and brown genes from the other insects were used to construct the evolutionary tree. Phylogenetic tree were constructed using the Neighbor-Joining method with MEGA (ver. 11.0.13) [20]. Numbers in the tree indicate bootstrap values (1000 replicates).
Line 92-93. “The forward primer (Table S1) comprised a T7 promoter… The reverse primer contained a partial sgRNA scaffold…”
In Table S1, for the sgRNA template primers it would be helpful to the reader to demarcate which parts of the sequence represent the T7 promoter, the sgRNA sequence and the sgRNA scaffold respectively. This could be done using bold font and/or different colored fonts, with a footnote to clarify the different features.
Response: According to the suggestion, we have underlined the T7 promoter, bold font marked gRNA, and italic the sgRNA scaffold in Table S1, also mark it in the footnote.
Line 102. “Cas9”
Is this referring to Cas9 protein or Cas9 mRNA? Please clarify here.
Response: According to the suggestion, we have replaced “cas9” with “cas9 protein” in line 106.
Line 106-107. “no further tagging was necessary.”
What is meant by tagging? This is not clear.
Response: We have replaced “Since the eggs were arranged neatly in 2 rows on the leaves, no further tagging was necessary.” to “Since the eggs of the A. hygrophila are laid in two neat rows on the leaves of A. philoxeroides (Figure S1), there is no need to align the eggs neatly to facilitate microinjection” in line 110.
Line 120-121. “the PCR products spanning all sgRNA target sites were amplified using specific primers”
Information about PCR thermocycling conditions is needed here.
Response: According to the suggestion, we have added the PCR thermocycling conditions: “The PCR conditions consisted of 95℃ for 3 min, followed by 35 cycles of 95℃ for 15 s, 55℃ for 15 s, and 72℃ for 2 min 20 s; a final extension step at 72℃ for 5 min, and held at 4℃.” in line 126.
Results.
Line 192-193. “A total of 300 eggs (G0) were injected with CRISPR-Cas9 ribonucleoprotein mixtures containing sgRNA1 and sgRNA2.”
For sake of clarity, please indicate if sgRNA1 and sgRNA2 were injected together in one mixture with Cas9, or if these were injected in separate RNP mixtures.
Response: We have replaced “A total of 300 eggs (G0) were injected with CRISPR-Cas9 ribonucleoprotein (RNP) mixtures containing sgRNA1 and sgRNA2 targeting AhW exon 3 and exon 5 (Figure 1A), respectively.” to “sgRNA1 and sgRNA2 are respectively located in exons 3 and 5 of the AhW. A total of 300 eggs (G0) were injected with RNP, consisting of Cas9 protein, sgRNA1 and sgRNA2.” in line 199.
Line 200. “genomes”
Replace with genomic DNA
Response: According to the suggestion, we have replaced “genomes” to “genomic DNA” in line 207.
Line 231. “AhW-KO_m 1”
Please verify and correct this as appropriate. In Figure 5, the only -1/+2 indel is seen for KO-5, not KO-1, which shows the 800bp deletion.
Response: Thanks to pointed out the error. We have replaced “AhW-KO_m 1” to “AhW-KO5” in line 239.
Line 239-240. “the wild-type AhW strain and the AhW-KO strain”
This is confusing, to refer to the wild type strain as AhW and the mutant strain as AhW-KO. Please consider to call the wild-type strain something more distinct.
Response: According to the suggestion, we have used "AhW strain" to refer to the wild type strain and "AhW-KO" to refer to the mutant strain throughout the manuscript.
Line 255-256. “and remained black during the rest of the pupal stage and throughout adult life (Figure 6)”
Does Figure 6 show the adult stage? The figure legend for Figure 6 says only pupal eyes, in that case, it’s not clear how anything can be mentioned about the adult stage here.
Response: Thanks to pointed out the error. We have replaced “They turned light red on the second day, gradually turned black on the third and fourth days, and remained black during the rest of the pupal stage and throughout adult life (Figure 6).” to “They turned light red on the second day, gradually turned black on the third and fourth days, and remained black during the rest of the pupal stage (Figure 6) and throughout adult life (Figure 3B).” in line 261
Discussion.
Line 283-284. “homologues of the white genes”
It is unclear what is being referred to here in the plural. Isn’t the target gene in this study only one homologue of one white gene?
Response: Thanks to pointed out the error. There was only one White in the A. hygrophila. We have replaced “homologues of the white genes” to “homologue of the white gene” in line 290.
Line 285-286.
“We characterized the mutant phenotype of these genes”
Again, here, isn’t it only one gene that you characterized the mutant phenotype for?
Response: Thanks to pointed out the error. We have replaced “We characterized the mutant phenotypes of these genes in A. hygrophila using the CRISPR/Cas9 system.” to “We characterized the mutant phenotype of this gene in A. hygrophila using the CRISPR/Cas9 system.” in line 292.
Line 288. “high rates of germline transformation”
From lines 192-194, it was indicated that 300 embryos were injected, from this, 8 adults were obtained, and only one showed evidence of the mutant phenotype. Considering these results, it is not clear what is meant by high rates of germline transformation.
Response: So far, from the references published in most other insects, the positive rate of gene editing by embryo injection of RNP is about 10-20%. Even in Drosophila melanogaster, when using transgenic methods, the efficiencies of ZFNs and TALEN were both less than 10% (PNAS, 2008, 105 (50): 19821-19826; Nucleic Acids Res, 2013, 41(17): e163). In contrast, by using the direct embryo injection method, the efficiency over 10% is already satisfactory.
Line 317. “As transgenic materials obtained from studies of Drosophila”
The construction of this sentence is confusing. What is meant by “as transgenic materials obtained from studies of Drosophila”? What transgenic materials have been obtained for this manuscript?
As transgenic materials obtained from studies of Drosophila [12-14], the white gene and white mutants can be used as selectable markers for RMCE systems and provide useful genetic materials for the study of A. hygrophila.
Response: Thanks to pointed out the error. We have replaced “As transgenic materials obtained from studies of Drosophila [12-14], the white gene and white mutants can be used as selectable markers for RMCE systems and provide useful genetic materials for the study of A. hygrophila.” to “the white gene and white mutants can be used as selectable markers for RMCE systems and provide useful genetic materials for the study of A. hygrophila such as in Drosophila [12-14].” in line 323.
Line 329. “We have established a highly efficient gene knockout system”
Referring back to my previous comment about the high rates of germline transmission, I think it is premature to say that it is highly efficient in this species based on the data provided here. It would be more accurate to simply say that you confirmed the efficacy of CRISPR-CAS9 in this species, A. hygrophila. This is a very nice result, but it should be overstated at this point without the corresponding data.
Response: According to our data, we got one positive mutant in 8 adults. For the other genes of A. hygrophila (data not published), increasing the number of eggs injected to 600 can produce sufficient mutant offspring. This efficiency can yield more than three independent mutant strains, which meeting the requirements of conventional gene function research.
Figures and Tables.
Figure 1. Since the scarlet and brown genes are included here in this figure, there should be some mention of these genes in the introduction, most appropriately where the white gene is being described.
Response: According to the suggestion, we have added "White forms heterodimers with brown (bw) and scarlet (st) respectively, which are responsible for the transport of pteridine precursors and ommochromes. The bw, st mutant have brown [11]and scarlet eyes [12].” in line 41.
Figure 3. For Figure 3B, it is not clear what we are looking at. Does the AhW and AhW-KO headers for 3A also apply to 3B?
Response: Yes, headers AhW and AhW-KO apply to 3A and 3B.
Furthermore, in the figure legend, on line 214, it mentioned “and germline-based G1 progeny (B). However, in the figure for the 3B panels, it says G2 male and G2 female, not G1 insects.
Response: Thanks to pointed out the error. We have replaced “Somatic CRISPR edits of G0 adults (A) and germline-based G1progeny (B).” to “Somatic CRISPR edits of G0 adults (A) and germline-based G2 progeny (B).” in line 222.
Reviewer 3 Report
Comments and Suggestions for Authors
Dear authors,
The article entitled CRISPR/Cas9-mediated Knockout of the White Gene in 2
Agasicles hygrophila represents a great contribution to the scientific field.
Before recommending it for publication, there are several details to be considered:
- Make sure that the full name Agasicles hygrophila appears only the first time it is mentioned; afterward, use the abbreviation.
- In the introduction section, you should also describe several studies in which the same method has been used on Agasicles hygrophila.
- Make sure that the gene names are italicized.
Author Response
The article entitled CRISPR/Cas9-mediated Knockout of the White Gene in Agasicles hygrophila represents a great contribution to the scientific field. Before recommending it for publication, there are several details to be considered:
Make sure that the full name Agasicles hygrophila appears only the first time it is mentioned; afterward, use the abbreviation.
Response: Thanks for pointing out the error, all the mistakes in the manuscript have been corrected.
In the introduction section, you should also describe several studies in which the same method has been used on Agasicles hygrophila.
Response: This article is the first one on gene editing in A. hygrophila, and there are no other reports.
Make sure that the gene names are italicized.
Response: The italics of the gene names have been checked.
Round 2
Reviewer 1 Report
Comments and Suggestions for Authors
-The authors have addressed all my concerns.